# Genetic Characterization of Human Rabies Vaccine Strain in Japan and Rabies Viruses Related to Vaccine Development from 1940s to 1980s

**DOI:** 10.3390/v14102152

**Published:** 2022-09-29

**Authors:** Madoka Horiya, Guillermo Posadas-Herrera, Mutsuyo Takayama-Ito, Yukie Yamaguchi, Itoe Iizuka-Shiota, Hirofumi Kato, Aikou Okamoto, Masayuki Saijo, Chang-Kweng Lim

**Affiliations:** 1Department of Virology I, National Institute of Infectious Diseases, 1-23-1 Toyama, Shinjuku, Tokyo 162-8640, Japan; 2Department of Obstetrics and Gynecology, The Jikei University School of Medicine, 3-25-8 Nishi-Shinbashi, Minato-ku, Tokyo 105-8461, Japan

**Keywords:** rabies vaccine, PCEC-KMB, rabies virus HEP-Flury, phylogenetic analyses, Lagos virus, Oita virus, rabies virus CVS, Rhabdoviridae, Lyssavirus, Ledantevirus

## Abstract

The rabies virus is widely distributed and vaccines are an important strategy to prevent its spread. The whole-genome sequences of rabies strains in relation to vaccine development provide essential information to maintain vaccine quality and develop new vaccines. However, the genetic characteristics of the purified chick embryo cell culture rabies vaccine, KM Biologics (PCECV-KMB), developed in Japan in the 1970s, have not been explored. In this study, we conducted a genome-wide analysis of the open reading frame regions of rabies strains discovered from the 1940s–1980s and used to develop chick embryo cell-adapted HEP-Flury small plaque-forming (CEF-S) strain, which is a vaccine strain of PCECV-KMB. The genetic characteristic of CEF-S, developed by acclimation of the HEP-Flury-NIID strain to one-day eggs and subsequently to chick embryo cells, were confirmed by comparing the genome identity and revealing the nine amino acid mutations between CEF-S and HEP-Flury-NIID. The efficacy of PCECV-KMB was evaluated using attack strains isolated in Thailand in the 1960s–1970s during vaccine development. Phylogenetic analyses of the attack strains classified them in the same Asian clade as the 2000s imported cases from the Philippines to Japan, suggesting that PCECV-KMB is adequate for preventing the spread of the current rabies virus.

## 1. Introduction

Rabies is a severe zoonotic disease that causes over 59,000 deaths annually and is especially prevalent in Asia and Africa. Current rabies vaccine development has led to a global initiative by the World Health Organization (WHO) to achieve zero human deaths from dog-mediated rabies by 2030 [1,2]. No domestic rabies outbreaks have been reported in Japan for over 50 years, except for four imported cases, one case from Nepal in 1970, two from the Philippines in 2006, and one from the Philippines in 2020 [3,4,5]. The last domestic reports were in 1956 and 1957 (in a human and a cat, respectively) due to the success of quarantine, vaccination, and geographical factors that separated Japan from other countries by sea [6,7]. However, Japan is surrounded by countries in which rabies is endemic, such as China, Korea, and Russia. China had the second-highest incidence of rabies after India, with 108,412 human rabies cases recorded between 1950 and 2004 [8,9,10]. South Korea, qualified as a rabies-free country by the WHO in 1984, experienced a reemergence of rabies on its border with North Korea in 1993. In 2002, 78 cases, including those in humans, were reported [11]. Thus, although a successful quarantine system and dog control have been maintained in Japan, the possibility of a rabies epidemic remains; and rabies vaccines play a crucial role.

Original rabies vaccines were derived from animal nerves or brain tissues; nerve tissue vaccine (NTV). The first reported rabies vaccination with an NTV occurred in 1885 when Louis Pasteur successfully treated a young boy bitten by a rabid dog [12]. The Pasteur-type vaccine may induce severe side effects because it contains myelin from nerve tissues [13]. Subsequently, a manufacturing method that reduces or eliminates nerve tissue has been developed using embryonated eggs and cell culture. Since the 1960s, human diploid cell vaccines (HDCV) and Vero cell culture-based vaccines have been used [12]. Two such vaccines, the PCECV-KM Biologics (PCECV-KMB) (KM Biologics Co., Ltd., Kumamoto, Japan) and Rabipur^®^ were approved in the 1980s [12,14,15,16]; these are immunogenic purified chicken embryo cell culture vaccines (PCECV) containing inactivated rabies virus.

In Japan, the Pasteur-type vaccine was used in humans between 1895 and the late 1940s [17]. The Nishigahara strain, developed in 1915 in Japan and derived from the original PV strain gifted by the Pasteur Institute in France, has been widely used as a Pasteur-type vaccine for humans [18]. After World War II, the Semple-type inactivated NTV, derived by inactivating the brain emulsion of goats infected with rabies virus with formalin, was introduced, resulting in the elimination of rabies from Japan by the end of the 1950s. Cell-cultured rabies vaccines were developed using chick embryo (CE) cells in the 1960s–1970s. Then, PCECV-KMB was approved in 1980 and has been widely used as a rabies vaccine in Japan ever since [14].

The genome sequence and the development procedure of the Rabipur vaccine have been widely studied; however, only a few studies have investigated the development procedures, diversity, and genetic stability of PCECV-KMB [12]. Furthermore, during PCECV-KMB development, various rabies strains were collected from endemic regions during the 1970s–1980s to evaluate the candidate vaccine in vivo. However, only partial genomic studies of rabies strains have been conducted [19,20]. RNA viruses are prone to mutations through replication within host cells. Therefore, vaccine strains and genome mutations during vaccine development provide critical information for the characterization and quality control of rabies vaccines. Competent authorities have tested the phenotypic stability of vaccines in vivo; however, genetic stability testing has been insufficient. Considering that vaccine management could be based on a seed lot system, genomic analysis of vaccine strains is an important research topic.

In this study, our objective was to elucidate the origin and development processes of PCECV-KMB. Next-generation sequencing (NGS) was used to perform genome-wide analyses of rabies virus strains previously obtained and established in our facility to develop cell culture-derived rabies vaccines for human use between the 1940s–1980s. A phylogenetic analysis was performed to compare the genetic and amino acid information between the parental and vaccine strains and other wild strains, which provided further information on the development and quality control of next-generation rabies vaccines. The data obtained in this study may help in the development of new vaccines and the implementation of seed lot systems for rabies vaccine quality control.

## 2. Materials and Methods

### 2.1. Viruses and Cell Culture

In this study, 20 strains of rabies virus, one Lagos Bat virus (LBV), and one Oita virus were investigated. Of the 18 rabies strains, the HEP-Flury one-day-egg passage (HFOP) strain, LEP-Flury one-day-egg passage (FOP) strain, chick embryo cell-adapted HEP-Flury large plaque-forming (CEF-L) strain, chick embryo cell-adapted HEP-Flury small plaque-forming (CEF-S) strain, challenge virus strain (CVS)-54, CVS-54 one-day-egg passage (COP) strain, M512, MDH [21], Nishigahara-NIID, one-day egg with Nishigahara-O30-M (NOPM), Takamen, ST1007, ST1009, ST1013, AY-163, Abha7, BK-3S, BK-3B, and LBV strain 8619NGA-NIID, which were lyophilized and stored at −80 °C, were directly used for genetic analysis. We propagated two rabies strains, HEP-Flury-NIID and LEP-Flury-NIID, from the stocks using Neuro-2A (NA) cells in Dulbecco’s modified Eagle’s medium (DMEM, Sigma, Saint Louis, MO, USA), supplemented with 2% calf serum and penicillin-streptomycin (Gibco, Waltham, MA, USA). The cultured viruses were twice passaged from the original brain samples into NA cells. The samples were clarified by centrifugation at 1000× *g* for 10 min to remove cell debris. The virus samples in the tissue culture supernatant were concentrated using Amicon Ultra centrifugal filter units (Millipore, Burlington, MA, USA) and stored at −80 °C until further use. Table 1 summarizes the ancestry and passaging history of the rabies strains analyzed. We used the Oita virus sequence as an outgroup for the phylogenetic analysis. The Oita virus classified genus Ledantevirus, family *Rhabdoviridae*, was isolated from the blood sample of a wild horseshoe bat, *Rhinolophus cornutus*, and stored at −80 °C as a 10%, suckling mouse-brain homogenate in Eagle’s minimum essential medium [22].

### 2.2. RNA Extraction

All freeze-dried virus stocks were reconstituted with PBS (-) (Thermo Fisher Scientific, Sunnyvale, CA, USA). Viral RNA was extracted directly using TRIzol LS (Life Technologies, Darmstadt, Germany) following the manufacturer’s instructions, resuspended in molecular grade distilled water, and stored at −80 °C until further use.

### 2.3. cDNA Library Amplification and Sequencing

Double-stranded cDNA was generated using the PrimeScript™ double-strand cDNA synthesis kit (Takara, Shiga, Japan). To prepare full-length cDNA clones and parental isolates, cDNA was fragmented using a sonicator (Bioruptor UCD-200, Diaginode, Liège, Belgium) and transformed into barcoded sequencing libraries using an Ion Plus Fragment Library Kit (Thermo Fisher Scientific) according to the manufacturer’s instructions. The resulting libraries were size selected using solid library size selection gels (Thermo Fisher Scientific), quantified using the Ion Library Quantification Kit (Thermo Fisher Scientific), and prepared for equimolar multiplexed sequencing using the Ion316 Chip Kit v2 on the Ion PGM system (Thermo Fisher Scientific) according to the manufacturer’s instructions.

An Ion PFM Hi-Q sequencing kit (Thermo Fisher Scientific) was used for template preparation and sequencing. Sequenced reads were quality checked and trimmed using Ion Torrent Suite version 5.0.4 software and assembled into contigs, via de novo assembly, using the CLC Genomic Workbench (Qiagen, Hilden, Germany). The contigs were compared to the GenBank non-redundant nucleotide database (BLASTn) to determine the contigs representing the rabies virus genome segments.

### 2.4. Sequence Alignment and Phylogenetic Analysis

Whole-genome sequences of rabies viruses were obtained from GenBank and combined with the 20 newly sequenced rabies virus samples to create the dataset used in this study (Appendix A). The LBV-8619NGA-NIID virus sequenced in this study was an outgroup. Alignment of the nucleotide and deduced amino acid sequences was performed using MEGA10.1.6 software [23]. The sequences of the N, P, M, G, and L genes of the amplified rabies virus were individually analyzed using Genetyx software version 19 (Genetyx, Tokyo, Japan). The complete available N, P, M, G, and L genes representing the main rabies virus lineages were obtained from GenBank (Appendix A). Each gene/region was aligned with amplified rabies virus gene sequences using MEGA10 software. Consensus phylogenetic trees were generated for the N, P, M, G, and L genes using MEGA10 software using the neighbor-joining method with 1000 bootstrap replicates.

## 3. Results

### 3.1. Next-Generation Sequencing Analysis

Fixated rabies strains related to PCECV-KMB and other rabies viruses collected for vaccine development between the 1940s and 1980s were sequenced and analyzed using NGS. Viral RNA was extracted directly from 10% brain emulsion reconstituted from freeze-dried stocks and/or culture supernatants. The genomes of the 20 rabies virus strains, one LBV, and one Oita virus sequenced in this study were deposited in GenBank (Table 1). All strains were analyzed using NGS and ORF region sequences were obtained. Regarding terminal sequence analysis, the regions that could be sequenced varied for each strain [24].

### 3.2. Phylogenetic Analysis of the Entire Genome of PCECV-KMB and Related Fixed Rabies Strains during the Development of the Rabies Vaccine against Reported Rabies Strains in Japan

Phylogenetic analysis was performed on the full-length genome of 79 RABV strains from the database and 21 newly sequenced strains (Table 1 and Appendix A). A phylogenetic analysis using MEGA10 software with the NJ algorithm and a consensus of 1000 bootstrap replicates revealed two major phylogenetic groups corresponding to bat- and dog-related RABVs, which were further subdivided into several clades (Figure 1), consistent with previously reported analyses of smaller datasets and individual RABV genes [25,26,27].

The bat-related group comprised two major clades: the bat RABV clade that circulates in the Americas and the other clade that includes viruses from American skunks and raccoons (RAC-SK). The bat RABVs clade comprised numerous strains isolated in the Americas between 1997 and 2012, including those isolated from cattle and dogs attacked by bats. Similarly, the dog-related group included six clades previously identified as Africa-2, Africa-3, Arctic-related, Asian, Cosmopolitan, and Indian subcontinent clades [28].

Phylogenetic analysis of the five concatenated genes (N, P, M, G, and L) distinguished various subclades and lineages among the Cosmopolitan clade, which was divided into five subclades (I–V). Cosmopolitan subclade I included the Nishigahara-NIID strain, NOPM, Takamen, PV-1965, and rabies strains isolated from various regions, including Africa, Europe, and China. The Cosmopolitan subclade II included a virus isolated from a Brazilian patient. Cosmopolitan Subclade III included several vaccine strains, including MDH, Pitman-Moore (PM), CVS-11 strains, Flury, HEP-Flury-NIID, HFOP, CEF-L, and CEF-S. Although CVS-11 and PM might have been derived from the original PV strain, the nucleotide sequence of the original PV strain was unknown because the current PV strain (PV-1965) is the PV strain transferred from Paris to Argentina, where it was maintained and subsequently stored back in the Pasteur Institute in Paris in 1965 [29]. The PV-1965 strain is included in Cosmopolitan subclade I, and the Nishigahara strain (derived from the original PV strain and stored in Japan) is also included in Cosmopolitan subclade I, suggesting that PM and CVS-11 were widely mutated during their development. The identity of the PV-1965 strain was 90.32% for PM and 90.15% for CVS-11. The Cosmopolitan subclade IV included the M512 strain, obtained from the same cluster as the isolated strain from the North American skunk (CASK2 strain). The Cosmopolitan subclade V included a strain isolated from Israeli dogs during the 1950s.

The Arctic and Arctic-related clades included strains isolated between 2007 and 2013 from Pakistan, India, Russia, Canada, and North Korea. The Komatsugawa strain isolated in the 1950s in Japan was also included in the Arctic-related clade. The Asian clade included ST1009, ST1013, ST1017, Abha7, BK-3B, BK-3S, and AY-163. These strains were used as attack strains in the PCECV-KMB evaluation. The Asian clade included the Chinese vaccine strain, which are strains isolated in southern China and Taiwan in the 2000s, and two strains isolated from patients with rabies strains imported from the Philippines to Japan in 2006. The Africa-3 clade included strains circulating in southern Africa that are phylogenetically distinct from other major RABV clades, particularly those circulating in Africa.

### 3.3. Phylogenetic Analysis of the Whole Genome and the N, P, M, G, and L Genes of Rabies Viruses

Phylogenetic tree analyses of the 20 rabies viruses sequenced in this study were performed using the whole genome, N, P, M, G, and L genes (Figure 2A–E). All phylogenetic analyses resulted in four clades (Figure 2A). The PCECV-KMB-related strains within the vaccine development group and LEP-Flury, as well as CVS-54 and MDH, belonged to the Cosmopolitan Subclade III. In comparison, M512 was included in the Cosmopolitan subclade IV. The Nishigahara-NIID strain, the NOPM strain, and the Takamen strain formed the Cosmopolitan subclade I. The identity between the Takamen and Nishigahara-NIID strains was as high as 99.76% (Table 2).

The isolates ST1009, 1013, 1017, Abha7, BK-3B, BK-3S, and AY-163, obtained from dogs and cats in Thailand, formed the Asian clade. The topology of the phylogenetic tree constructed for each gene (N, P, M, G, and L) was like that obtained for the whole-genome sequences (Figure 2B–F) [25].

### 3.4. Identity and Nucleotide Differences of Vaccine Strain CEF-S against Parent Strain HEP-Flury-NIID and Related Strains

The identities of the vaccine strain CEF-S and the parental strains are shown in Table 3. The identity between CEF-S and HFOP was 99.77%, whereas that between CEF-S and HEP-Flury-NIID was 99.66%. A comparison of amino acid mutations between HFOP and CEF-S revealed two mutations in the P region (residue 28, I to V and residue 289, D to N), one mutation in the M region (residue 16, T to I), two mutations in the G region (residue 164, V to E and residue 259, N to D), and three mutations in the L region (residue 745, Q to R; residue 1319, R to K; and residue 1848, R to K). In contrast, amino acid mutations between HEP-Flury-NIID and HFOP revealed one mutation in the M region (residue 105, A to V), two mutations in the G region (residue 206, T to I and residue 504, R to I), and one mutation in the L region (residue 383, L to S) (Figure 3A, Appendix A). CEF-S shared 99.79% identity with CEF-L. PCECV-KMB was developed from the CEF-S adapted to CE cells from the HFOP strain, forming small plaques. CEF-L was obtained by isolating a viral strain that formed large plaques using the same process as that used for CEF-S. CEF- S has several unique amino acid mutations compared to four other related strains, at residue 289 in the P region (D to N), residue 18 in the M region (T to I), and residue 745 and residue 1848 in the L region (Q to R and R to K, respectively) (Figure 3A, Appendix A), which may be responsible for the smaller plaque phenotype.

### 3.5. Nucleotide and Amino Acid Differences between CVS-54 and Other CVS Strains

The CVS-54 strain is used to assess the potency of the human rabies vaccine at the National Control Laboratory, Japan. Therefore, it is valuable to compare its identity with the international challenge virus standard, CVS-11. We compared the complete sequence of the CVS-54 strain with that of the COP, CVS-11, and CVS-N2c strains to determine their identity and the putative variation in the amino acid sequence (Figure 3D) [30,31]. CVS-54 showed 99.95% identity with CVS-11, with one amino acid mutation in the P region and one in the G region. Glycoprotein mutations were not observed at antigenic sites I (residues 231), II (residues 34–42, 198–200), III (residues 330–338), IV (residue 264), a (residue 342), or other regions believed to be involved in antigenicity or immunogenicity. The presence of arginine or lysine at position 333 in the glycoprotein was associated with rabies virus pathogenicity. Glutamine, leucine, or glycine at position 333 of the glycoprotein significantly reduces the virulence of the rabies virus [32,33,34]. The arginine in position 333 of the glycoprotein in CVS-54 maintains the high virulence of this strain (Appendix A) [32,33]. This confirms that CVS-54 was well preserved despite being introduced to Japan over 70 years ago.

### 3.6. Nucleotide Differences between the Nishigahara-NIID Strain and the Takamen Strain

Nishigahara is a PV-derived strain long used in Japanese Pasteur-type vaccines. The NOPM strain was produced by adapting the Nishigahara-NIID strain to one-day eggs. These two strains and the Takamen strain isolated in Japan, and converted to a fixed strain, were similar in all phylogenetic analyses (Figure 1 and Figure 2). Therefore, the nucleotide sequence identity and amino acid variations of these strains were evaluated (Table 2, Figure 3C, Appendix A). The Nishigahara-NIID strain had a higher identity (98.95%) with the Takamen strain than with the NOPM strain. Seventeen nucleotide mutations were discovered between the Nishigahara-NIID and Takamen strains with only two amino acid mutations at residues 37 and 385 in the G region (N to D and I to T, respectively), and three at residues 237, 1592, and 1767 in the L region (L to F, Q to R, and K to R, respectively). Focusing on the antigenic site of the glycoprotein, mutations were observed at residues 37 and 40, which correspond to the glycoprotein antigenic site II, in the strains Nishigahara, NOPM, and Takamen. Dietzschold et al. reported that the peptide containing amino acids 34 through 42 were joined by a disulfide bridge with the peptide containing amino acids 198 through 200 [35,36,37].

## 4. Discussion

In this study, rabies virus strains related to PCECV-KMB development and evaluation were analyzed using NGS to confirm and evaluate the origin and development processes of PCECV-KMB. To date, phylogenetic analyses have been conducted primarily on viral genes such as the N and P genes; however, only a few studies have evaluated the full length of viral genomes [38,39]. These phylogenetic analyses have been performed on relatively few sequences derived from specific regions or animal hosts [25]. Furthermore, most phylogenetic analyses of whole genomes have been performed primarily on street strains, Furthermore, most phylogenetic analyses of whole genomes have been performed primarily on street strains, and few analyses have included fixed strains such as vaccine strains. Therefore, in this study, we performed phylogenetic analyses of vaccine strains from various regions using full-length genomes and including strains isolated from several types of host animals.

Numerous vaccine strains were derived from the original Louis Pasteur isolate from the suburbs of Paris in 1882 [29]. However, the current PV strain (PV-1965), whose full-length sequence was reported in 1988 [40], was transferred to the Pasteur Institute in 1965 from the Institute of Zoonoses Louis Pasteur, Argentina, and was probably derived from the original PV strain [29].

In Japan, the Pasteur-type vaccine for humans had been in use from 1895 until the late 1940s [18,41,42]. The Japanese Pasteur-type rabies vaccines primarily use the original PV and Nishigahara strains. The original PV strain first introduced in Japan in 1883 was from the Pasteur Institute, and the Nishigahara strain was derived from the original PV strain in 1915 through an alternate subculture in guinea pigs and rabbits for 90 generations; subsequently, the culture period was shortened to three days and was repurposed for human use [18,41]. Surprisingly, the identity between the PV-1965 and Nishigahara-NIID strains was only 91.0%, indicating that both strains had drastically altered their identity during maintenance in animal models (Table 1).

Attempts have been made to improve rabies vaccines due to the safety and potency of Pasteur-type vaccines [42]. In the 1940s, marzonin and ultraviolet-inactivated vaccines developed from the Nishigahara strain were considered for human use. In 1953, the Semple-type vaccine, based on the Nishigahara and MDH strains, was approved. Although the incubation period of the MDH strain to obtain high viral titer was longer, and the neutralizing antibody titers induced by the MDH strain were lower than the Nishigahara strain, both vaccines were successful [42]. However, they were like the original Pasteur-type vaccine, which contained high levels of myelin, causing sensitization in some vaccine recipients and, in extreme cases, fatal encephalitis. Alternatives to this approach include producing an inactivated rabies vaccine using embryonated eggs infected with the rabies virus [43] or suckling mouse brains with a lower level of myelin than the adult mouse brain [44]. In the 1960s, Japanese scientists attempted to develop a vaccine using embryonated one-day eggs. The HEP-Flury, LEP-Flury, CVS-54, and Nishigahara strains were acclimated to one-day egg chick embryos, and the HFOP, FOP, COP, and NOPM strains were established. The HFOP strain exhibits the fastest growth rate among the one-day egg-acclimated strains [43,45,46,47]. NOPM-based beta-propiolactone-inactivated one-day egg candidate vaccines have advanced to a stage where their safety in human phase trials has been confirmed because of their immunogenicity [48,49]. In 1973, an inactivated fluenzalida-type Nishigahara strain-based vaccine was approved for use in Japan. However, these approaches are not free of components derived from nerve tissue or autoimmune reactions.

The PCECV-KMB was developed in 1965 and approved in 1980 in Japan [14,50]. PCECV-KMB is a lyophilized product prepared by inactivating the viral suspension of CEF-S fixed with β-Propiolactone, followed by concentrating and purifying it via ultracentrifugation (Figure 4) [14,50,51]. Other strains, such as Nishigahara-NIID, CVS-54, and M512, were adapted to CE cells; however, titers were not sufficiently high for vaccine production [50]. The Flury strain was isolated from human patients with rabies in the USA in 1939 via passage through the chicken brain and subsequently adapted to chick embryos. The HEP-Flury strain, which had been derived from the Flury strain cultivated for over 130 generations using chicken embryos, was first reported in 1954. The HEP-Flury-NIID strain was supplied by the Pasteur Institute and stored in a lyophilized state after the 200th to 210th passages in embryos [47]. The LEP-Flury-NIID strains analyzed in this study were gifted by Prof. Koprowski (Center for Neurovirology Jefferson Cancer Institute) after 75 generations of eggs [45]. CEF-S was established through plaque purification by cloning small plaques on agar plates after the adaptation of the HFOP strain to CE cells. Using a serum-free medium, Kondo passed the HFOP strain over 100 times through 7-day-old CE cells [14,50,51]. A procedure similar to that used for the CEF-S strain was used to obtain the CEF-L strain by plaque purification of large plaques. A vaccine candidate containing CEF-S has been shown to induce higher neutralizing antibody titers than those prepared with CEF-L [14]. The genetic identity and amino acid mutations of CEF-S against HEP-Flury-NIID and its related derivatives confirmed the PCECV-KMB development procedure (Figure 3A).

We showed that the Nishigahara-NIID strain differed from the previously reported Nishigahara strain (AB044824) by seven amino acids (Figure 3C and Figure 4) and was not considered significantly different from the Nishigahara strain. The identity between the PV-1965 and Nishigahara-NIID strains was 91.0% (Table 2). The Takamen strain, a fixed Japanese street strain, was highly homologous to the Nishigahara-NIID strain (Table 2).

The Takamen strain was derived from the Takahashi strain, which was isolated from the brain of a 6-year-old girl bitten on her right cheek by a rabid dog in Tokyo in 1946. However, the veterinary vaccination history of the rabid dog remains unknown because the canine had escaped. The patient received neither pre- nor post-exposure prophylaxis. After a two-year incubation period, the patient died of rabies in 1948 [52]. The Takamen strain was isolated from the brain of a mouse that had been highly immunized intraperitoneally twice at a 4-day-interval with the live Takahashi strain obtained from four serial passages in the mouse brain [53]. The characteristics of the Takamen strain differ from the Takahashi strain distinctly in the short incubation period in rabbits and mice, the high intracerebral infectivity in mice, and the absence of subcutaneous infectivity in rabbits. Although these characteristics seem to show a fixation tendency, unlike other fixed strains, Negri bodies have been found in the brain of mice and rabbits from the 1st to the 54th and 13th passages, respectively, disappearing after that. Therefore, it was assumed that the Takamen strain was a viral strain that acquired characteristics very close to those of fixed strains and was fixed during a relatively short period through the passaging [54]. The Takamen strain was delegated to our facility in 1962 and was conserved in adult mice for 12 passages and in suckling mice for 12 passages before being stocked [53], the Nishigahara-NIID and Takamen strains showed high identity, with only six amino acid differences (Table 2 and Figure 3C). In the 1940s, the Nishigahara strain was used as a veterinary phenolized live vaccine for dogs [41]. The high identity between the Takamen and Nishigahara-NIID strains indicates a close relationship between Takamen and the veterinary phenolized live vaccine strain Nishigahara. 

The CVS-54 strain was transferred from the National Institutes of Health, USA (NIH) in 1949 and used in Japan as a challenge virus standard in vaccine potency assays [55,56]. The CVS-54 strain in NIID was obtained after 54 passages of the CVS strain through mouse brains and was adapted to NA-C1300 cells [45,55]. The COP strain is a viral strain obtained by adapting the CVS-54 strain to embryonated eggs for one day [45]. The CVS-54 strain showed a 99.95% identity with the CVS-11 strain, with only three amino acid mutations and well-conserved regions. These results indicate that CVS-54 is genetically identical to CVS-11 in determining the potency of rabies vaccines.

The MDH strain was supplied to Japan by the NIH in 1949 as a seed virus for the manufacture of inactivated vaccines and was stored at our facility [42,55]. In the phylogenetic analysis, the MDH strain was included in Cosmopolitan Subclade III, which comprised PM, and the Ethiopian vaccine strain EPH-VAC. The MDH and PM strains shared a 97.08% identity. The origin of the MDH strain has not been reported; however, our data indicate that MDH could be closely related to the PM strain [21].

The M512 strain was isolated from a spotted skunk in the USA, passaged four times through primary hamster kidney cell culture by Dr. Johnson (California State Department of Public Health), gifted to our facility in 1961 as a street strain, and adapted to primary CE cells [45,50,56]. M512 was included in Cosmopolitan subclade IV based on our phylogenetic analysis with street strains isolated from a skunk in California in 1974 and 1994, consistent with a previous report [45].

ST1009, ST1013, ST1017, BK-3B, and BK-3S are street strains collected in Thailand between 1977 and 1985 [19,57]. AY-163 is a fixed strain isolated in 1963 in Ayutthaya, Thailand, passaged in four-week-old mice, and cryopreserved in 1967 [19]. Abha7 was isolated in Ayutthaya, Thailand, in 1976 and was used to determine the efficacy of PCECV-KMB using a monkey model [58]. Our phylogenetic analysis showed that strains AY-163, ST1009, ST1013, ST1017, Abha7, BK-3B, and BK-3S belonged to the Asian clade, like strained 8743THA collected in Thailand in 1983, and strain Leo4 collected in Laos in 2011 [38,59]. 

The Kyoto and Yokohama strains isolated from rabies cases in Japan were imported from the Philippines and classified into the Asian clade [6,60]. The phylogenetic geographic patterns of the Asian clade are often maintained over time [28], indicating that the strains of the Asian clade did not experience drastic changes over the years within specific regions. In addition, individuals who were bitten by rabid dogs or suspected of having survived attacks were included in post-exposure prophylaxis studies using PCECV-KMB in Thailand in the 1970s [58]. Therefore, PCECV-KMB was effective in preventing endemic rabies.

In this study, we analyzed the entire ORF region of LBV NGA8619-NIID and Oita virus 296/1972 in NGS using the same method as rabies strains for use as an out-group. NGA8619-NIID was collected from a straw-colored fruit bat (*Eidolon helvum*) in Nigeria [38], transferred to our facility, and passed through a mouse brain. Since the first LBV was isolated from the brain of a Nigerian fruit bat in 1956 and was later identified as a rabies-related virus, several LBV infections have been reported. Two clusters were formed when the phylogenetic tree was analyzed using other LBV sequences (Figure 5). It was well conserved since 8619NGA-NIID was highly identified with the original 8619NGA. Oita virus 296/1972 was isolated from the blood of a wild horseshoe bat in 1972 [22]. The entire ORF region of Oita virus 296/1972 was analyzed in this study.

Only a few studies have analyzed the whole-genome sequence of LBV and Oita viruses. We used NGS to investigate the whole genome of LBV and Oita viruses and concluded that such a method may be useful for rhabdoviruses other than rabies virus [61].

## 5. Conclusions

Vaccines are the most effective prophylactic public health tool that facilitates the prevention of the spread of infectious diseases. Licensing and quality control require the determination of consensus genome sequences of replication-competent infectious agents in vaccines [62]. In this study, we analyzed the whole genomes of rabies strains relevant to vaccine development in Japan and showed that whole-genome analysis using NGS is a high-throughput method that can be applied to other lyssa viruses, including LBV. Based on these results, we reviewed the procedure for the development of a rabies vaccine in Japan and analyzed strains produced during vaccine development to provide useful information for vaccine quality control and future vaccine improvements. Recent improvements in sequencing technologies enabled highly reliable sequencing of complete genomes and genetic analysis of populations at a high resolution. The latter is significant for RNA viruses, which comprise fluctuating heterogeneous populations, rather than being genetically stable entities. Further accumulation of such data and their analysis can be crucial for the development of new vaccines. This information must be integrated into the existing regulatory framework; thus, challenging both licensing authorities and vaccine producers to develop new quality control criteria.

## Figures and Tables

**Figure 1 viruses-14-02152-f001:**
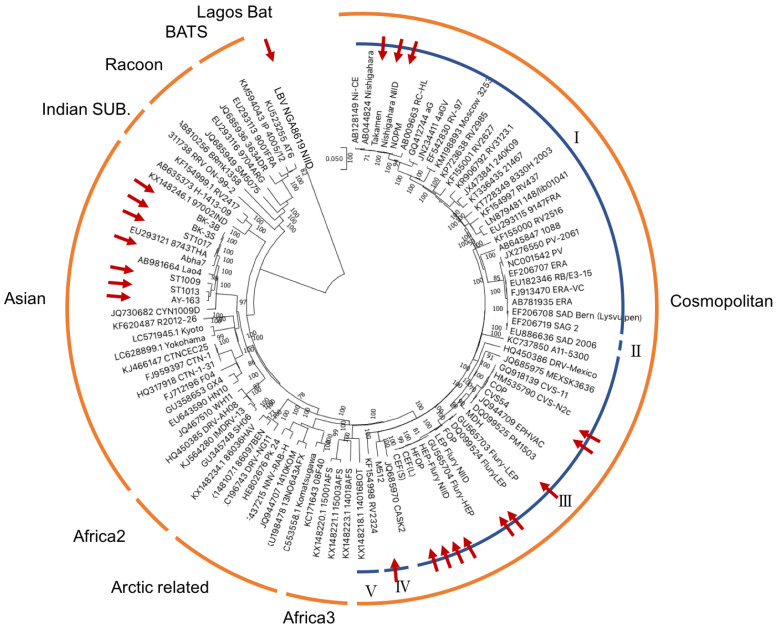
Phylogenetic analysis of rabies strains. Phylogenetic analysis of all coding sequences of 20 rabies strains (analyzed in this study), one Lagos Bat virus, and Gen Bank sequences was performed using the neighbor-joining method and 1000 bootstrap analyses (Mega10.1 software). Chip names are represented by virus names and GenBank accession numbers. Posterior probabilities between 70 and 100% are indicated by numbers. Posterior probabilities below 70% are not indicated. The strains analyzed in this study are marked with red arrows. LBV 8619NGA-NIID was the outgroup. The tree is rooted at the midpoint for clarity and is divided into bat-associated RABVs, including raccoon-skunk clades and bat clades, and dog-associated RABVs, including Africa-2, Arctic-associated, Asian, Indian, and Cosmopolitan clades. The Cosmopolitan clade is divided into five subclades, named Cosmopolitan subclades I–V.

**Figure 2 viruses-14-02152-f002:**
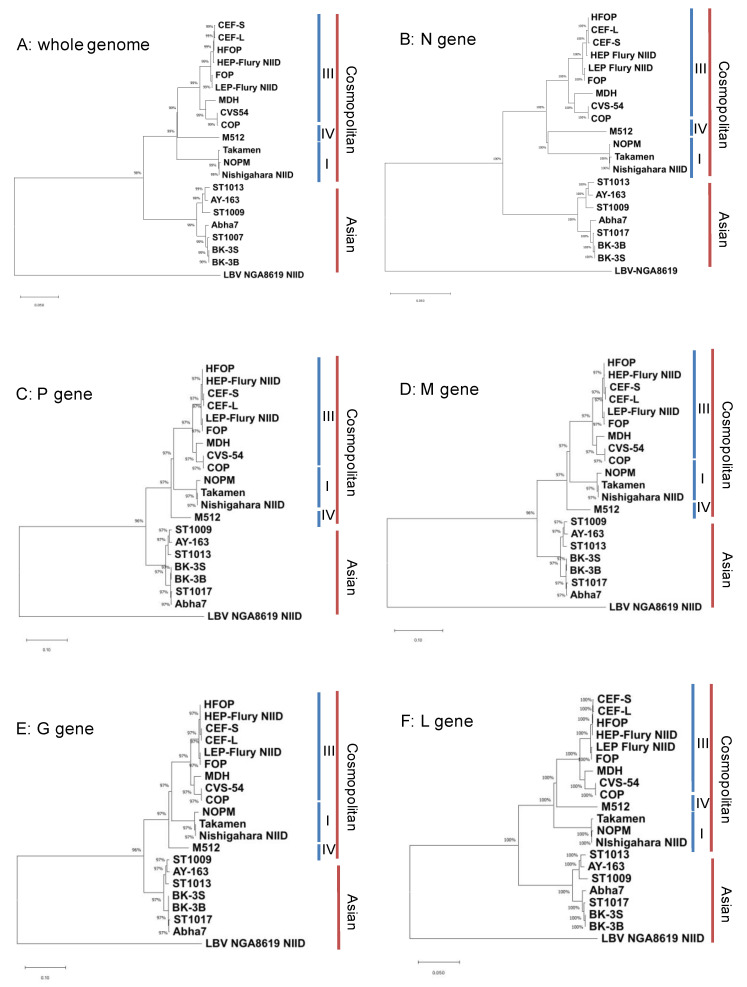
Phylogenetic analysis of the rabies strains sequenced in this study. Phylogenetic analysis of the rabies strains sequenced in this study was performed for the whole-genome sequence and the N, P, M, G, and L genes using the neighbor-joining method and 1000 bootstrap analyses (Mega10.1 software). Posterior probabilities between 90 and 100% are indicated by numbers. The clades on the right side of the tree correspond to the clades shown in Figure 1. (**A**) Whole genome, (**B**) N, (**C**) P, (**D**) M, (**E**) G, and (**F**) L genes. The division was the same for each gene and for the whole sequence.

**Figure 3 viruses-14-02152-f003:**
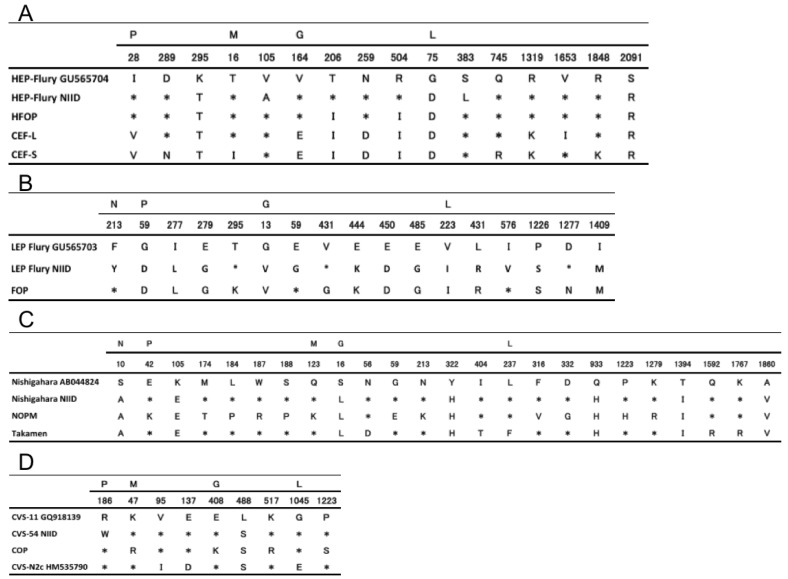
Comparison of the deduced amino acid sequence of rabies viruses. We compared the deduced amino acid sequences of (**A**) HEP-Flury-related strain, (**B**) LEP-Flury-related strain, (**C**), Nishigahara-related strain and Takemen strain, and (**D**) CVS-related strain. The first row shows the sequences of the HEP-Flury strain (GU565704), LEP-Flury (GU565703), Nishigahara (AB044824), and CVS-11 (GQ918139) strains with a one-letter symbol. For the other strains, only nonidentical amino acids are depicted at each corresponding position of the first-row strain, and identity is depicted as *.

**Figure 4 viruses-14-02152-f004:**
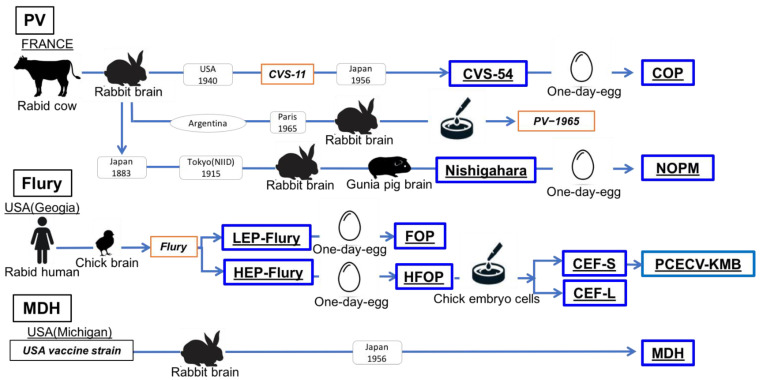
Schematic representation of the lineage of fixed strains used as vaccine seeds and analyzed in this study, as well as the other relevant strains. The strains sequenced in this study are highlighted. Strains not analyzed in this study are listed in italics.

**Figure 5 viruses-14-02152-f005:**
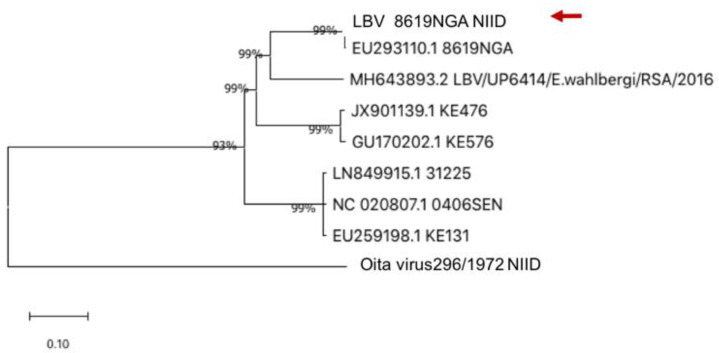
Phylogenetic analysis of LBV strains. Phylogenetic analysis of Lagos Bat strains using Mega10.1 software with the neighbor-joining method and 1000 bootstrap analyses. Phylogenetic analysis was performed using the LBV-8619NGA-NIID strain and other Lagos Bat viral strains that have been analyzed and deposited in GenBank. The Oita virus was the outgroup.

**Table 1 viruses-14-02152-t001:** List of virus strains analyzed in this study.

Strain	Accession Number	Host/Parental Strain	
HEP-Flury-NIID	LC717409	Flury	
HFOP	LC717410	HEP-Flury	The HEP-Flury strain adapted to one-day eggs
CEF-L	LC717411	HFOP	The HFOP adapted to chick embryo fibroblast
CEF-S	LC717412	HFOP	The HFOP adapted to chick embryo fibroblast
LEP-Flury-NIID	LC717413	Flury	
FOP	LC717414	LEP-Flury	The LEP-Flury strain adapted to one-day eggs
Nishigahara-NIID	LC717415	PV	Pasteur-type Vaccine Strain in Japan
NOPM	LC717416	Nishigahara-NIID	The Nishigahara strain adapted to one-day-egg
CVS-54	LC717417	PV	Obtained from the US NIH in 1949
COP	LC717418	CVS-54	The CVS-54 strain adapted one-day-eggs
Takamen	LC717419	Takahashi	Fixed from the Takahashi strain isolated from the human case in 1948 in Japan
MDH	LC717420	Unknown	Obtained from the US NIH in 1949
M512	LC717421	Spotted skunk	Isolated in the US in the 1960s
AY-163	LC717422	Dog	Isolated in Ayutthaya, Thailand, 1963 mouse brain passage
ST 1009	LC717423	Dog	Isolated in Bangkok, Thailand, 1985 mouse brain passage
ST 1013	LC717424	Dog	Isolated in Bangkok, Thailand, 1985 mouse brain passage
ST 1017	LC717425	Dog	Isolated in Bangkok, Thailand, 1985 mouse brain passage
Abha7	LC717426	Dog	Isolated in Bangkok, Thailand, 1977 mouse brain passage
BK-3B	LC717427	Dog	Isolated in Bangkok, Thailand, 1977 mouse brain passage
BK-3S	LC717428	Cat	Isolated in Bangkok, Thailand, 1977 mouse brain passage
Lagos bat virus 8619NGA-NIID	LC717429	Bat	Lagos bat virus Isolated in Nigeria, 1956 mouse brain passage
Oita virus 296/1972-NIID	LC717430	Bat	Oita virus isolated in Japan, 1972

**Table 2 viruses-14-02152-t002:** Identity of Nishigahara-NIID, NOPM, and Takamen strains.

		NOPM [%]	Takamen [%]
Nishigahara-NIID	Full	98.35	98.95
	N	100.00	99.78
	P	98.32	100.00
	M	99.53	100.00
	G	99.62	99.62
	L	99.90	99.78
NOPM	Full		97.86
	N		99.78
	P		98.32
	M		99.53
	G		99.24
	L		99.66

**Table 3 viruses-14-02152-t003:** Identity of PCECV-KMB-related strains.

		CEF-L [%]	HFOP [%]	HEP-Flury-NIID [%]
CEF-S	Full	99.79	99.77	99.66
	N	100.00	100.00	99.63
	P	99.66	99.33	99.44
	M	99.53	99.53	99.18
	G	99.43	99.05	99.43
	L	99.86	99.86	99.83
CEF-L	Full		99.76	99.67
	N		100.00	99.78
	P		99.66	99.55
	M		100.00	99.51
	G		99.43	99.49
	L		99.91	99.80
HFOP	Full			99.86
	N			99.78
	P			99.89
	M			99.67
	G			99.75
	L			99.92

## Data Availability

Not applicable.

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
