# Peer review of "Genetic Characterization of Human Rabies Vaccine Strain in Japan and Rabies Viruses Related to Vaccine Development from 1940s to 1980s"

_viruses, 2022, doi:10.3390/v14102152_

Round 1
Reviewer 1 Report
This paper describes the origin of PCECV-KMB, which Japan has produced since 1980 as the only rabies vaccine for human use, along with the origins of other rabies-fixed strains that were nominated between 1940 and 1980. This paper describes in detail the similarities and differences with other isolates reported around the world by analyzing the sequence using WGS. Japan has been a completely free country with no domestic outbreaks of rabies not only in humans but also in animals for the past 70 years. However, the production of the Japanese-made PCECV-KMB vaccine was suspended in 2019, and currently the WHO prequalified PCECV (Rabipur) is in circulation. Under these circumstances, in order for Japan to secure its own seeds for human rabies vaccines and to regain the ability to produce a stable and uniform vaccine domestically, various viruses that have been accumulated so far will be used. It is considered important to determine the genetic information of the strain. From the above points, this paper is considered to have important significance that leads to vaccine measures.
Here are some comments on some of the concerns. 1. L66: “Semple-type” appears for the first time and needs explanation or introduction. 2. L138: 21 Newly sequenced, but Table 1 has 22 virus strains. which one is correct? 3. L175-181: Also, PV-1965 appears for the first time and requires explanation or introduction. 4. L245-249: Description in text does not match Fig. 3A data
5. LBV-8619NGA-NIID and Oita-virus 296/1972-NIID are not related to this theme, so I do not feel the need to describe them.
Author Response
A point-by-point response to Reviewer 1
Thank you for reviewing our manuscript. Below are our point-by-point responses to the comments and suggestions.
General comment and suggestions
This paper describes the origin of PCECV-KMB, which Japan has produced since 1980 as the only rabies vaccine for human use, along with the origins of other rabies-fixed strains that were nominated between 1940 and 1980. This paper describes in detail the similarities and differences with other isolates reported around the world by analyzing the sequence using NGS. Japan has been a completely free country with no domestic outbreaks of rabies not only in humans but also in animals for the past 70 years. However, the production of the Japanese-made PCECV-KMB vaccine was suspended in 2019, and currently the WHO prequalified PCECV (Rabipur) is in circulation. Under these circumstances, in order for Japan to secure its own seeds for human rabies vaccines and to regain the ability to produce a stable and uniform vaccine domestically, various viruses that have been accumulated so far will be used. It is considered important to determine the genetic information of the strain. From the above points, this paper is considered to have important significance that leads to vaccine measures.
Response: Thank you very much for reviewing our manuscript and offering your valuable advice. We have addressed your comments with point-by-point responses and revised the manuscript accordingly.
- L66: “Semple-type” appears for the first time and needs explanation or introduction.
Response: Thank you very much for your excellent suggestion. According to your excellent suggestion, we added an explanation of this term in the revised manuscript (lines 64–66).
- L138: 21 Newly sequenced, but Table 1 has 22 virus strains. which one is correct?
Response: Thank you for your very thoughtful remarks. In this study, we analyzed 22 virus strains, including 20 rabies viruses, Lagos bat virus, and Oita virus. This is why the numbers look different depending on the description. I have followed your suggestion and added an explanation of the number of strains analyzed to make it easier for readers to understand (lines 93-94, 103, 140, and 155–156).
- L175-181: Also, PV-1965 appears for the first time and requires explanation or introduction.
Response: Thank you for your valuable suggestions. I have followed your suggestion and have added an explanation regarding PV-1965 in the Results section (lines 181–188).
- L245-249: Description in text does not match Fig. 3A data
Response: I appreciate your pertinent and accurate remarks. We have reviewed the data and replaced Figure 3A. Thank you very much for your careful reading of our manuscript (Figure 3A).
- LBV-8619NGA-NIID and Oita-virus 296/1972-NIID are not related to this theme, so I do not feel the need to describe them.
Response: Thank you for your valuable suggestions. As per your suggestion, we have deleted Section 3.7 on Page 7 and appropriately revised the Discussion. One of the reasons we analyzed the Lagos bat virus and Oita virus was to demonstrate that our NGS method is versatile, and can be used not only for the rabies virus but also for other rhabdoviruses. Based on this, we have revised the manuscript (lines 439–442)

Reviewer 2 Report
This study by Dr. Lim's team is very interesting. Although "history is always a mystery", a few findings in the investigation remind us of the accuracy of "records" as revealed by the cutting-edge technology of NGS data.
1. It is believed that the Nishigahara strain was from the original PV strain. As mentioned in the manuscript, detailed on page 10, lines 313-318, "... the Nishigahara strain was derived from the original PV strain in 1915 through alternate subculture in guinea pigs and rabbits... the identity between the PV-1965 and Nishigahara-VIID was only 91.0%...".
2. On page 5, lines 181-182, the identity of PV-1965 versus PM is 90.32%, and identity of PV-1965 versus CVS-11 is 90.15%.
3. On page 11, lines 364-366, "Takamen strain, a rabid dog virus in Tokyo in 1946, a fixed Japanese street strain, was highly homologous to the Nishigahara -NIID strain (98.95-100%, Table 3).
Our questions are:
Q1: is the PV-1965 the original PV from Paris? Since in the study, the PM and CVS-11 were compared to the PV-1965. Without "record" for the support of the PV-1965, it should be treated as an "assumption".
Q2: the stunning high identity between Takamen strain and Nishigahara-NIID strain (98.95-100%) could be more convincingly explained as a "mislabel" of the virus other than the differences in virus passages.
We appreciate the great efforts the authors put into this investigation, which is of great interest and help us in understanding of the virus history, but our limitations are always with the "records". Vaccines are evaluated by safety, efficacy and administration, and the NGS data are supplementary.
Other minor questions:
MQ1: "Oita virus was isolated rom the blood sample of a wild horseshoe bat (22)" on page 4, lines 112-114. It would be interesting to know how rabies virus survives and transmits in blood, since rabies virus is highly neurotrophic.
MQ2: Section 3.7 on page 7 could be omitted. The study is mainly on vaccine strains, and Lagos bat viruses are not the topic.
Author Response
A point-by-point response to Reviewer 2
Thank you for reviewing our manuscript. Below are our point-by-point responses to the comments and suggestions.
General comment and suggestions
This study by Dr. Lim's team is very interesting. Although "history is always a mystery", a few findings in the investigation remind us of the accuracy of "records" as revealed by the cutting-edge technology of NGS data.
- It is believed that the Nishigahara strain was from the original PV strain. As mentioned in the manuscript, detailed on page 10, lines 313-318, "... the Nishigahara strain was derived from the original PV strain in 1915 through alternate subculture in guineapigs and rabbits... the identity between the PV-1965 and Nishigahara-VIID was only 91.0%...".
- On page 5, lines 181-182, the identity of PV-1965 versus PM is 90.32%, and identity of PV-1965 versus CVS-11 is 90.15%.
- On page 11, lines 364-366, "Takamen strain, a rabid dog virus in Tokyo in 1946, a fixed Japanese street strain, was highly homologous to the Nishigahara -NIID strain (98.95-100%, Table3).
Response: Thank you very much for reviewing our manuscript and offering valuable advice. We have addressed your comments with point-by-point responses and revised the manuscript accordingly.
Q1: is the PV-1965 the original PV from Paris? Since in the study, the PM and CVS-11 were compared to the PV-1965. Without "record" for the support of the PV-1965, it should be treated as an "assumption".
Response: Thank you for your thought-provoking remarks. There is a record of the course of PV-1965 in which the original PV strain was delegated from the Pasteur Institute to Argentina and then again to the Pasteur Institute. The process of the transfer of this virus strain was described in the Discussion section (lines 307–314). However, in response to your suggestion, we have added an explanation of this transfer to the description of PV-1965 in Result (lines 181–188).
Q2: the stunning high identity between Takamen strain and Nishigahara-NIID strain (98.95-100%) could be more convincingly explained as a "mislabel" of the virus other than the differences in virus passages.
Response: Thank you so much for your very useful suggestions. In the manuscript, we revised the additional explanation of the characterization and isolation process of Takamen strains (lines 377–392). The Takamen strain was unexpectedly isolated from the brain of a mouse that had been highly immunized intraperitoneally twice with live Takahashi strain. Surprisingly, the characteristics of the Takamen strain clearly differ from those of the Takahashi strain: a short incubation period in rabbits and mice, high intracerebral infectivity in mice, and lack of subcutaneous infectivity in rabbits. These characteristics appear to indicate a fixation tendency from the beginning of its isolation, but unlike other fixed strains, Negri bodies were found in mouse and rabbit brains until the 54th and 13th passages from the beginning and disappeared after that (detailed in reference 54, added in this revision: line 602). Thus, the Takamen strain was isolated after careful inoculation and passaging. Based on the references and our results, it was assumed that the Takamen strain was a viral strain that acquired unique characteristics very close to those of fixed strains and was fixed during a relatively short period through the passaging. Thus, we considered that the high identity between the Takamen strain and the Nishigahara-NIID strain may indicate some important phenomenon. In addition, when Arai et al. previously analyzed the N-protein sequence of the Takamen strain transferred to our laboratory, they also showed high identity between the Nishigahara-NIID strain and the Takamen strains, which is consistent with our results (detailed in references 19 and 20), and on the basis of these arguments, we do not believe that the sample was mislabeled.
MQ1: "Oita virus was isolated rom the blood sample of a wild horseshoe bat (22)" on page 4, lines 112-114. It would be interesting to know how rabies virus survives and transmits in blood, since rabies virus is highly neurotrophic.
Response: Thank you for raising these interesting points. According to Iwasaki et al. (reference 22) who reported Oita virus 296/1972, classified genus Ledantevirus, family Rhabdoviridae, it is morphologically similar to lyssavirus (lines 93-94, 113, and 155-156). However, the virus was isolated from bat blood. Ephemeroviruses and vesicloviruses cause hematogenous infection (viremia), whereas lyssaviruses are believed and demonstrated to cause neurogenic infection (viremia). In this regard, the presence of this virus in bat blood suggests that it spread via bat blood. However, immunohistological analysis detected no viral antigens in extra-neuronal tissues after subcutaneous or intraperitoneal inoculation of milk-drinking mice, suggesting that Oita virus 296/1972 is a neurotropic virus (line 113). Thus, although the virus is inconsistent regarding the route of infection, the details of the virus still seem to be unknown.
MQ2: Section 3.7 on page 7 could be omitted. The study is mainly on vaccine strains, and Lagos bat viruses are not the topic.
Response: Thank you for your valuable suggestions. As per your suggestion, we have deleted section 3.7 on page 7 and revised the “Discussion” section. One of the reasons we analyzed the Lagos bat virus and Oita virus was to demonstrate that our NGS method is versatile, and can be used not only for rabies virus but also for other rhabdoviruses. Based on this, we have revised the relevant section of the manuscript (lines 439–442).
